# Evaluating the Real-World Predictive Utility of Karnofsky and ECOG Performance Status for 90-Day Survival After Oncologic Surgery for Metastatic Spinal Tumors

**DOI:** 10.3390/cancers17223629

**Published:** 2025-11-12

**Authors:** Rafael De La Garza Ramos, Ali Haider Bangash, Sertac Kirnaz, Rose Fluss, Victoria Cao, Alexander Alexandrov, Liza Belman, Saikiran G. Murthy, Yaroslav Gelfand, Reza Yassari

**Affiliations:** 1Spine Tumor Mechanics and Outcomes Research (TUMOR) Lab, Montefiore Medical Center, Albert Einstein College of Medicine, Bronx, NY 10467, USA; alhaider@montefiore.org (A.H.B.); skirnaz@montefiore.org (S.K.); victoria.cao@einsteinmed.edu (V.C.); alexander.alexandrov@einsteinmed.edu (A.A.); liza.belman@einsteinmed.edu (L.B.); samurthy@montefiore.org (S.G.M.); ygelfand@montefiore.org (Y.G.); ryassari@montefiore.org (R.Y.); 2Department of Neurological Surgery, Montefiore Medical Center, Albert Einstein College of Medicine, Bronx, NY 10467, USA

**Keywords:** metastatic spine disease, metastatic spine tumor surgery, Karnofsky performance score, Eastern Cooperative Oncology Group performance status, 90-day survival, discrimination, calibration, decision curve analysis, surgical decision-making, patient selection

## Abstract

When cancer spreads to the spine, surgery can help maintain quality of life, but doctors need reliable ways to identify which patients might benefit most. Performance status is a measure of a patient’s physical abilities and daily functioning. It is widely used to help predict which patients might survive long enough after surgery to justify the procedure. However, it remains unclear how accurately these scores actually predict short-term survival when used alone. Our study evaluated two common performance status measures to determine how well they predict whether patients will survive at least 90 days after spine surgery for cancer that has spread to the spine. We found that while poor performance status was associated with worse survival, neither scale performed particularly well at predicting individual outcomes when used alone, especially in the equivocal survival probability range. These findings suggest surgeons should use caution when relying heavily on performance status alone for surgical decision-making in these vulnerable patients.

## 1. Introduction

Metastatic spine tumor surgery (MSTS) plays an important role in maintaining neurologic function, spinal stability, and quality of life [1,2,3,4]. As systemic therapies continue to improve survival in many cancer subtypes, careful patient selection for surgery is of paramount importance. Performance status, most commonly measured by the Karnofsky Performance Status (KPS) or the Eastern Cooperative Oncology Group (ECOG) scale, is an important factor that is very frequently cited as an independent predictor of survival [5,6,7,8]. It is considered an important variable when assessing surgical candidacy based on management guidelines for patients with metastatic spine disease (MSD), which recommend a minimum expected survival of 90 days [9]. Likewise, it is commonly used to assess eligibility for clinical trial enrollment in cancer patients [10].

Nevertheless, while the independent statistical association between poor baseline performance status and shorter postoperative survival is well documented, less is known about the actual real-world predictive utility of KPS and ECOG performance status (ECOG-PS) when used alone. Prior studies have focused primarily on multivariable models and hazard ratios in their analyses [6,11,12,13], but there is limited data on how well performance status scores discriminate between survivors and non-survivors or whether they are sufficiently calibrated to predict short-term outcome [14]. Furthermore, even though many prognostic tools for MSD have incorporated performance status as a component [15,16], including the revised Tokuhashi score (which explicitly uses KPS for “general condition” scoring) [13,17] and the Bayesian decision-support tool PathFx that incorporates performance status (KPS or ECOG) [18], they have demonstrated poor prognostication of short-term outcomes for patients with MSD who experienced worsened survival [19]. This underscores that the clinical utility of performance status for informing surgical decision-making in this patient population remains uncertain.

Therefore, the objective of this study was to evaluate the predictive utility of KPS and ECOG-PS scores for 90-day survival after MSTS by assessing the discrimination, calibration, and clinical utility of both scores. Our goal was to determine whether performance status alone could be sufficient to predict early postoperative survival in this patient population.

## 2. Materials and Methods

### 2.1. Study Design and Data Source

This study was approved by our Institutional Review Board (IRB 2024-16293). This is a retrospective review of a prospectively maintained institutional surgical spinal oncology database.

A total of 234 adult patients operated on for MSD between April 2012 and April 2025 were assessed for eligibility. Inclusion criteria included patients with (1) complete preoperative KPS and ECOG-PS score data and (2) at least 90-day follow-up (or less if they had documented death). Our study selection algorithm is depicted in Figure 1. Based on this, a total of 175 patients were included in our final analytic sample.

Indications for surgery included metastatic spinal cord compression (MSCC) and/or spinal instability. Patients underwent a combination of surgical decompression, stabilization, and postoperative radiation therapy on a case-by-case basis and as determined by the surgeon and radiation oncologist.

### 2.2. Examined Variables

The following data were extracted from our spinal oncology surgical database: age at surgery, sex, body mass index, American Association of Anesthesiologists (ASA) class, preoperative serum albumin level, prognostic nutritional index, neurological status at presentation (Frankel D–E vs. A–C), non-ambulatory status at presentation, de novo metastatic disease, primary tumor type, presence of MSCC, and presence of a pathological vertebral compression fracture.

Preoperative performance status, assessed by the KPS and ECOG-PS scores, was ascertained from preoperative medical oncology and/or radiation oncology notes and referred to the performance status of the patient within 30 days before surgery. The KPS ranges from 0 to 100 in 10-point increments, with a score of 0 representing death and higher scores indicating better overall performance status. The system is based on a patient’s ability to carry out activities of daily living and the degree of assistance needed. On the other hand, ECOG-PS ranges from 0 to 5 in 1-point increments, with a score of 0 denoting a patient being fully active without restriction and a score of 5 meaning “death”. The score is based on a patient’s ability to carry out activities of daily living and also incorporates ambulatory status.

### 2.3. Study Endpoint

The primary study endpoint was postoperative 90-day survival.

### 2.4. Statistical Analyses

Exploratory and descriptive analyses were first performed. Univariable logistic regression was used to assess the association between performance status and 90-day survival, with KPS and ECOG-PS scores entered as independent continuous variables. Results were reported as odds ratios (ORs) with 95% confidence intervals (CIs).

The predictive performance of KPS and ECOG-PS was then evaluated across four domains: discrimination, diagnostic accuracy, calibration, and clinical utility. Discrimination was assessed by calculating the area under the receiver operating characteristic curve (AUC) for KPS and ECOG-PS and comparing them using DeLong’s test. An AUC ≥ 0.7 was deemed clinically useful. Diagnostic accuracy was assessed by calculating the sensitivity, specificity, positive predictive value (PPV), and negative predictive value (NPV) for multiple cutoff values. Predicted probabilities for 90-day survival were derived from the univariable logistic regression models using KPS and ECOG-PS scores as continuous predictors. These probabilities were used to assess model calibration and to calculate net benefit in decision curve analysis (DCA). Calibration at the 90-day timepoint was evaluated using calibration plots, which compared predicted vs. observed survival across deciles of risk, and by calculating the Brier score (the average squared difference between predicted risk and observed risk), ranging from 0 (perfect) to 1 (worst). Lastly, clinical utility was assessed using DCA, which estimates the net benefit of each model across a range of threshold probabilities, relative to “treat all” and “treat none” strategies. The threshold probability represents the minimum predicted probability of survival at which a clinician would choose to offer surgery. Net benefit incorporates both true positives and false positives while reflecting whether a model improves decision-making compared to default (“treat all” and “treat none”) strategies. This approach allows for evaluation of whether a predictive tool offers added clinical value at decision points relevant to patient care. Statistical significance was defined as a *p*-value < 0.05. All statistical analyses were conducted using STATA (StataNow/BE version 19.5) and Python (version 3.10) with Scikit-learn (version 1.3), Matplotlib (version 3.7), and Statsmodels (version 0.14) libraries.

## 3. Results

A total of 175 patients were analyzed, with a mean age of 62 (±13) years and 59% (103 of 175) male sex (Table 1). The median ASA class was 3. The most common primary tumor types were prostate (20%, n = 35), lung (15%, n = 27), breast (15%, n = 26), and hematologic malignancies (19%, n = 34). At presentation, 28% (n = 48) of patients were non-ambulatory, and 89% (n = 155) had MSCC. Pathologic vertebral compression fractures were present in 52% (n = 90) of cases. The distribution of both KPS and ECOG-PS scores is shown in Figure 2.

The 90-day survival rate was 73% (127 of 175). On univariable analysis, increasing KPS was associated with higher odds of 90-day survival (OR 1.02 [95% CI 1.01 to 1.05]; *p* = 0.001). Increasing ECOG-PS was associated with lower odds of 90-day survival (OR 0.51 [95% CI 0.36 to 0.73]; *p* < 0.001). Predicted probabilities, used for both the calibration and DCA analyses below, are shown in Figure 3.

### 3.1. Discrimination

KPS achieved an AUC of 0.65 (95% CI: 0.56–0.75), compared to 0.68 (95% CI: 0.59–0.77) for ECOG-PS. The difference between the two was not statistically significant (ΔC = 0.03, *p* = 0.46) (Figure 4).

### 3.2. Diagnostic Accuracy

Across both performance status measures, only reasonable diagnostic accuracy was achieved at intermediate cutoffs (Table 2). For KPS, a threshold of ≥70 offered the most even compromise across sensitivity (0.66), specificity (0.62), PPV (0.82), and NPV (0.41), whereas for ECOG-PS, a cutoff of ≤2 achieved a similar balance (sensitivity = 0.76, specificity = 0.5, PPV = 0.8, NPV = 0.44).

### 3.3. Calibration

Calibration plots for both scores demonstrated reasonable agreement between predicted and observed 90-day survival across deciles. Brier scores were 0.185 for KPS and 0.182 for ECOG-PS, indicating modest but comparable overall calibration performance (Figure 5).

### 3.4. Clinical Utility

DCA for 90-day survival showed that both KPS and ECOG-PS scores provided minimal net benefit across a range of threshold probabilities (Figure 5). While ECOG-PS demonstrated slightly higher net benefit than KPS at intermediate thresholds (30–60%), the difference was small (0.5 for ECOG-PS compared to 0.495 for KPS), with both scores reaching a maximum net benefit of 0.608. At a clinically relevant threshold of 50% predicted survival, the net benefit was 0.474 for ECOG and 0.451 for KPS. Neither score consistently outperformed the “treat all” or “treat none” strategies across most thresholds.

## 4. Discussion

In this study, we evaluated the utility of KPS and ECOG-PS scores for predicting 90-day survival following MSTS, a metric that is frequently used to determine surgical candidacy [2,4,9,20]. We found that both scores were associated with survival on univariable analysis but overall showed modest predictive utility, particularly at moderate performance status scores. By assessing discrimination, calibration, and decision-level utility, we provided a detailed evaluation of their performance in surgical decision-making for this vulnerable population.

### 4.1. Interpretation and Generalizability

Our results showed that both KPS and ECOG-PS scores were significantly associated with 90-day survival on regression analysis, but neither demonstrated strong discriminatory ability (AUC ≥ 0.7), with AUC values of 0.65 and 0.68, respectively. While prior studies have consistently identified performance status as an “independent predictor” of survival in patients with MSD [6,20,21], our findings show that statistical association (as oftentimes shown in multivariable models) equates to neither clinical utility nor discrimination at the individual patient level. In agreement with these findings, Jang RW et al. assessed the discriminative ability of KPS and ECOG-PS for overall survival in 1655 patients with advanced cancer, finding AUCs of 0.63 and 0.64, respectively [22].

Our diagnostic accuracy analysis revealed a fundamental challenge in operationalizing performance status for binary surgical decisions, with there being no single cutoff achieving both high sensitivity and specificity. At commonly cited thresholds (KPS ≥ 70, ECOG-PS ≤ 2), sensitivity ranged from 0.66 to 0.76, but specificity remained modest at 0.50–0.62, meaning that approximately half of patients who did not survive 90 days would have been classified as acceptable surgical candidates. Conversely, more stringent cutoffs (KPS ≥ 80, ECOG-PS ≤ 1) improved specificity to 0.69–0.77 but sacrificed sensitivity to 0.49–0.51, potentially excluding many patients who would have survived. This accuracy-coverage tradeoff could elucidate why consensus thresholds remain elusive despite substantial research [15,23,24]. Notably, performance status demonstrated the clearest utility at the extremes, with patients having KPS < 40 or ECOG-PS 4 experiencing a near-certain 90-day mortality (PPV ≈ 0.75 for mortality prediction), while those with ECOG-PS 0 had an 82% survival probability. However, these unambiguous categories represented a minority of patients in our cohort. The majority fell in intermediate ranges (KPS 40–80, ECOG-PS 1–3), where diagnostic accuracy was majorly inadequate for confident individual-level prediction. This pattern suggests performance status functions more effectively as a screening tool to identify patients at the margins of surgical candidacy rather than as a precise classifier for the majority falling in the equivocal middle range.

Calibration (assessing how well predicted probabilities reflect actual outcomes) was modest, with Brier scores of 0.185 for KPS and 0.182 for ECOG-PS. These scores are comparable to those reported in other oncologic prognostic models [16,18], suggesting that while predictions were not perfectly aligned with observed outcomes, they were directionally reasonable. However, perfect calibration is rare, particularly in models that include only a single predictor [25].

We used DCA to assess whether KPS or ECOG-PS could guide surgical decision-making. DCA estimates the net clinical benefit of a predictive model across a range of threshold probabilities, which represent the minimum predicted chance of survival at which a clinician might consider offering surgery. In our study cohort, very few patients survived when their predicted 90-day survival probability (based solely on performance status) was below 30%, and most survived when probabilities exceeded 60%. Thus, performance status alone classified outcomes reasonably well at these extremes. However, clinical decisions at these extremes are typically straightforward: patients with very poor performance status (ECOG 4 or KPS < 40) are less likely to be surgical candidates, while those with excellent function (ECOG 0 or KPS ≥ 90) are generally offered surgery. As clinicians already act with confidence in these settings, performance status added little incremental value. The greatest uncertainty, and therefore the greatest opportunity for model guidance, occurred between predicted probabilities of 30% and 60% in our cohort. Within that range, ECOG demonstrated slightly higher net benefit than KPS (average net benefit 0.5 vs. 0.495) and achieved a modestly higher value at a 50% threshold (0.474 vs. 0.451). Both models achieved maximal net benefit near a 60% predicted survival probability, generally corresponding to ECOG 0–1 or KPS ≥ 70. However, this range did not map directly to a specific ECOG or KPS score, limiting clinical interpretability. These findings further reinforce that while performance status is statistically associated with survival, it has limited standalone utility in guiding short-term surgical decision-making, particularly in the range where decisions are most difficult.

Although clinicians consider multiple factors, performance status remains among the most frequently cited predictors of survival for patients with MSD, implying a value that extends beyond statistical association. In a systematic review of 22 studies encompassing 7779 patients, Bollen L et al. found that performance status was one of the two prognostic factors most frequently associated with survival, reporting a 93% positive association [6]. In a similar investigation, Luksanapruska P et al. found that KPS < 70 and ECOG 3–4 were commonly reported independent poor prognostic factors [7]. However, a widely accepted cutoff exists for neither the KPS nor the ECOG-PS to predict 90-day survival for patients managed with MSTS [23,26,27,28,29,30,31]. Moreover, in reality, multiple other factors, including but not limited to tumor histology, frailty, extent of disease, and ambulatory status, play an important role in surgical decision-making [5]. Given this context, our findings provide objective data on the limitations of relying on KPS or ECOG-PS alone and underscore the value of comprehensive and patient-centered approaches to surgical decision-making.

### 4.2. Limitations

This study has several important limitations. First, it was a retrospective, single-institution analysis, which may limit the generalizability of our findings. Second, the use of univariable logistic regression models was intentional, as our goal was to isolate the predictive performance of performance status scores alone. However, in clinical practice, performance status is typically evaluated alongside other prognostic or predictive variables, which could improve model performance in multivariable settings. Third, although 90-day survival is a pragmatic endpoint, it may not capture longer-term outcomes such as ambulation or health-related quality of life. Furthermore, although performance status was assessed using validated scales, variability in documentation or interpretation could have introduced measurement bias. Additionally, while this study focused on short-term survival as a pragmatic and objective endpoint, it did not assess postoperative functional or quality-of-life benefit. Evaluating which patients derive meaningful improvement from MSTS, particularly across different performance status levels, represents an important next step. Future prospective studies should, therefore, incorporate functional recovery, pain relief, and health-related quality of life alongside survival to better define surgical benefit and optimize patient selection. Despite these limitations, however, our study provides a structured, clinically grounded assessment of performance status as a standalone predictor of early postoperative survival.

## 5. Conclusions

In this single-institution cohort of patients undergoing metastatic spine tumor surgery, preoperative KPS and ECOG-PS were statistically associated with 90-day survival and demonstrated clear clinical relevance at performance extremes. Patients with very high or very low performance status exhibited predictable survival patterns, supporting their utility in straightforward surgical decisions. However, overall predictive performance remained modest, with fair discrimination and calibration. Likewise, the decision-level utility was limited, particularly in the 30–60% probability range where clinical ambiguity is most common. These findings highlight the distinction between statistical association and actionable prediction and caution against overreliance on performance status alone for surgical selection, particularly for patients with moderate PS.

## Figures and Tables

**Figure 1 cancers-17-03629-f001:**
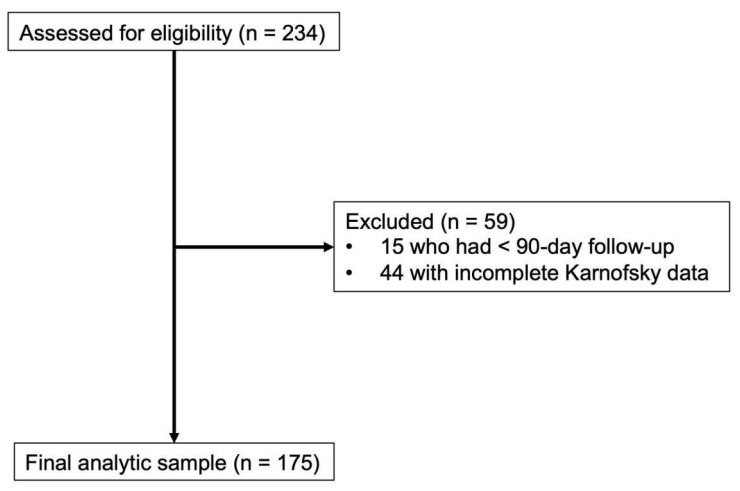
Patient selection algorithm for patients managed with metastatic spine tumor surgery. Of 234 patients managed between April 2012 and April 2025 who were assessed for eligibility, 59 patients were excluded due to less than 90-day follow-up (n = 15) and incomplete preoperative Karnofsky Performance Status data (n = 44). The final analytic sample comprised 175 patients who were included in the study.

**Figure 2 cancers-17-03629-f002:**
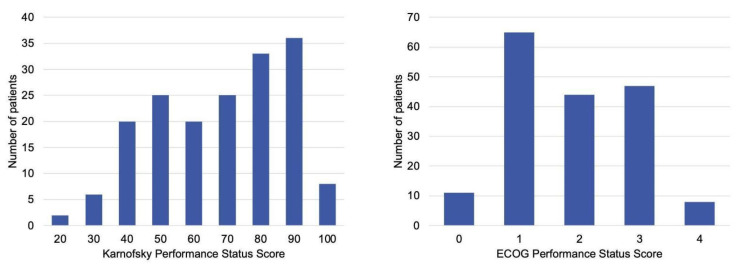
Baseline, preoperative performance status scores distribution of patients managed with metastatic spine tumor surgery. (**Left**) Karnofsky Performance Status scores, which range from 0 (death) to 100 (normal activity, no complaints). (**Right**) Eastern Cooperative Oncology Group Performance Status scores, which range from 0 (fully active) to 5 (death). Most patients presented with KPS ≥ 70 or ECOG ≤ 2, indicating generally preserved baseline function before surgery.

**Figure 3 cancers-17-03629-f003:**
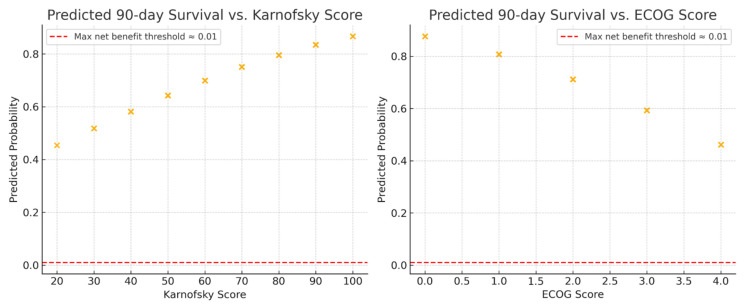
Predicted 90-day survival probabilities across functional performance scales for patients managed with metastatic spine tumor surgery. (**Left**) Relationship between predicted 90-day survival probability and Karnofsky Performance Status (KPS). (**Right**) Relationship between predicted 90-day survival probability and Eastern Cooperative Oncology Group Performance Status (ECOG-PS). Each point represents the model-predicted probability of survival derived from univariable logistic regression. Higher KPS and lower ECOG-PS scores were associated with increased likelihood of 90-day survival.

**Figure 4 cancers-17-03629-f004:**
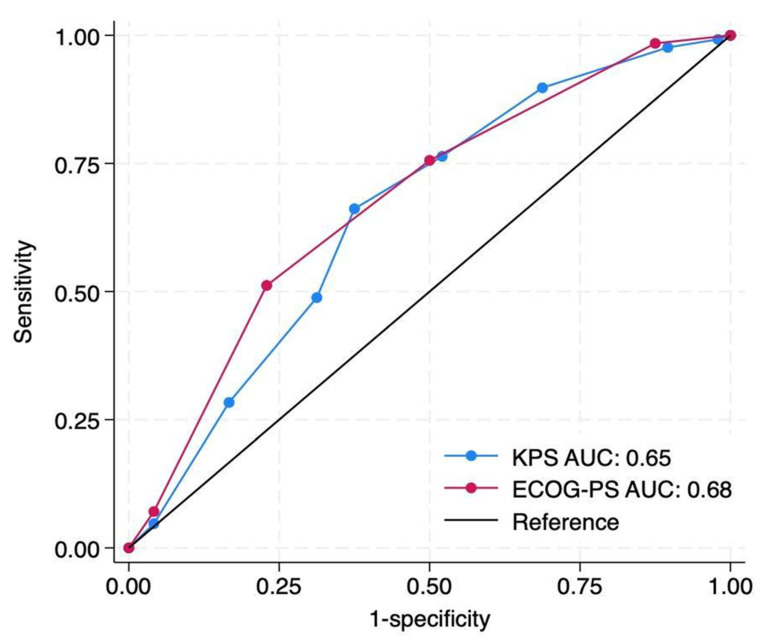
Discrimination comparison of baseline preoperative Karnofsky Performance Status (KPS) and Eastern Cooperative Oncology Group Performance Status (ECOG-PS) scores for prediction of 90-day survival after metastatic spine tumor surgery. Receiver operating characteristic curves demonstrate modest discrimination for both scales, with area under the curve (AUC) values of 0.65 for KPS *(blue)* and 0.68 for ECOG-PS *(red)*. The difference between AUCs was statistically insignificant (ΔAUC = 0.03; p = 0.46 by DeLong’s test). The diagonal black line represents the reference *(no-discrimination)* line.

**Figure 5 cancers-17-03629-f005:**
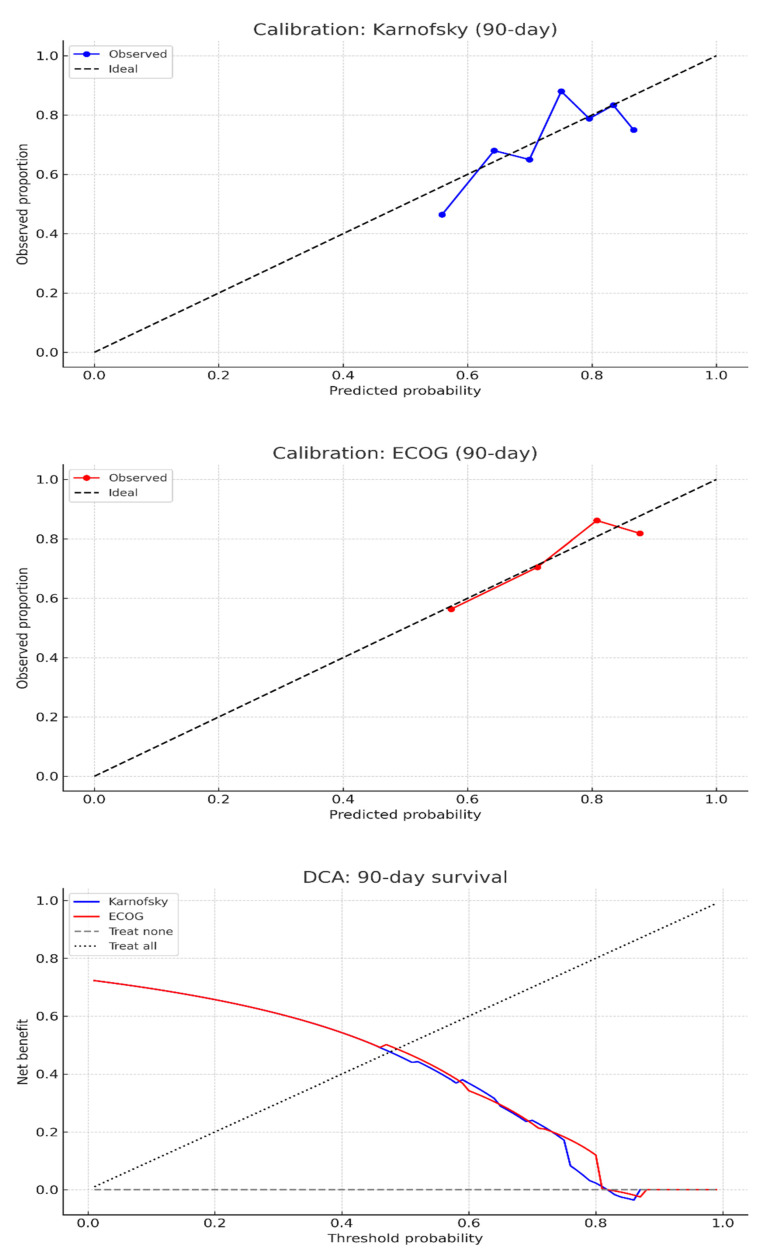
Calibration and decision curve analysis for 90-day survival prediction following metastatic spine tumor surgery using preoperative performance status scores. (**Top**) Calibration plots demonstrate reasonable agreement between predicted and observed 90-day survival probabilities for both the Karnofsky Performance Status *(blue)* and Eastern Cooperative Oncology Group Performance Status *(red)* scales, with points approximating the ideal reference line. (**Bottom**) Decision curve analysis indicates minimal net clinical benefit across most threshold probabilities for both models, suggesting limited utility of either scale for individualized decision-making after metastatic spine tumor surgery.

**Table 1 cancers-17-03629-t001:** Baseline and oncological characteristics of 175 patients managed with metastatic spine tumor surgery.

Parameter	Value, %
Age (mean, SD)	62, 13
Male sex (%)	103 (59)
BMI (mean kg/m^2^, SD)	25.7, 5.8
ASA class (median, IQR)	3 (3–3)
Albumin (mean, SD)	3.7 (0.6)
PNI (mean, SD)	42.4 (7.5)
Neurological status	
Frankel D–E (%)	139 (80)
Frankel A–C (%)	34 (20)
Non-ambulatory at presentation (%)	48 (28)
De novo metastasis (%)	64 (37)
Primary cancer	
Breast cancer (%)	26 (15)
Lung cancer (%)	27 (15)
Prostate cancer (%)	35 (20)
Colorectal cancer (%)	6 (3)
Kidney cancer (%)	6 (3)
Thyroid cancer (%)	4 (2)
Hematological cancer (%)	34 (19)
Other cancer (%)	37 (21)
Metastatic spinal cord compression	155 (89)
Pathologic vertebral compression fracture (%)	90 (52)

SD: standard deviation; ECOG: Eastern Cooperative Oncology Group; BMI: Body mass index.

**Table 2 cancers-17-03629-t002:** Diagnostic accuracy of Karnofsky Performance Status and Eastern Cooperative Oncology Group Performance Status scores for predicting 90-day survival for patients managed with metastatic spine tumor surgery.

Performance Score Thresholds	Sensitivity	Specificity	Positive Predictive Value	Negative Predictive Value
**KPS**				
≥20	1	0	0.73	0
≥30	0.99	0.02	0.73	0.5
≥40	0.98	0.1	0.74	0.62
≥50	0.9	0.31	0.78	0.54
≥60	0.76	0.48	0.8	0.43
≥70	0.66	0.62	0.82	0.41
≥80	0.49	0.69	0.81	0.34
≥90	0.28	0.83	0.82	0.31
≥100	0.05	0.96	0.75	0.28
**ECOG-PS**				
≤0	0.07	0.96	0.82	0.28
≤1	0.51	0.77	0.86	0.37
≤2	0.76	0.5	0.8	0.44
≤3	0.98	0.12	0.75	0.75
≤4	1	0	0.73	0

KPS: Karnofsky Performance Status; ECOG-PS: Eastern Cooperative Oncology Group Performance Status.

## Data Availability

All the data pertinent to this study are included in the article. Further inquiries can be directed to the corresponding author.

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
