# Peer review of "Evaluating the Real-World Predictive Utility of Karnofsky and ECOG Performance Status for 90-Day Survival After Oncologic Surgery for Metastatic Spinal Tumors"

_cancers, 2025, doi:10.3390/cancers17223629_

Round 1
Reviewer 1 Report
Comments and Suggestions for Authors
-
In the Introduction, it might be helpful to expand briefly on how existing prognostic systems (such as Tokuhashi, Tomita, and PathFx) incorporate performance status and why their predictive value for short-term survival remains limited. This would better highlight the novelty of the current analysis.
-
A few sentences in the Abstract and Discussion could be streamlined to improve clarity and flow, without altering the meaning.
-
Figure legends could include a bit more explanatory detail—perhaps mentioning confidence intervals or adding a short interpretive note—to make them more self-contained.
-
If possible, an exploratory multivariable or sensitivity analysis could provide additional perspective, even if it remains outside the main objective of the study.
Author Response
Title of the Manuscript: Evaluating the Real-World Predictive Utility of Karnofsky and ECOG Performance Status for 90-Day Survival After Oncologic Surgery for Metastatic Spinal Tumors
Manuscript Number: cancers-3955605
Reviewer 1
Comment 1: In the Introduction, it might be helpful to expand briefly on how existing prognostic systems (such as Tokuhashi, Tomita, and PathFx) incorporate performance status and why their predictive value for short-term survival remains limited. This would better highlight the novelty of the current analysis.
Authors’ Response: Thank you very much for this insightful comment. We fully agree that incorporating a brief discussion of how existing prognostic systems integrate performance status and their limitations for short-term survival would better contextualize the novelty of our analysis. Accordingly, we have revised the “Introduction” section to include this information and to clarify the rationale for our study.
Change to Text: “Furthermore, even though many prognostic tools for MSD have incorporated performance status as a component, including the revised Tokuhashi score (which explicitly uses KPS for "general condition" scoring), and the Bayesian decision-support tool PathFx that incorporates performance status (KPS or ECOG), they have demonstrated poor prognostication of short-term outcomes for patients with MSD who experienced worsened survival. This underscores that the clinical utility of performance status for informing surgical decision-making in this patient population remains uncertain.”
Comment 2: A few sentences in the Abstract and Discussion could be streamlined to improve clarity and flow, without altering the meaning.
Authors’ Response: Thank you for this helpful comment. We reviewed the “Abstract” and “Discussion” sections to identify opportunities to improve clarity and flow. Several sentences were streamlined by merging related ideas, reducing redundancy, and simplifying phrasing without altering meaning, as highlighted in yellow for your kind consideration. These edits enhance readability while preserving the technical accuracy and nuance of our findings.
Comment 3: Figure legends could include a bit more explanatory detail—perhaps mentioning confidence intervals or adding a short interpretive note—to make them more self-contained.
Authors’ Response: Thank you for this valuable comment. We carefully reviewed all figure legends to enhance clarity, completeness, and interpretability. Each legend was expanded to provide sufficient context and describe the key findings. These revisions ensure that each figure is self-contained and aligns with journal standards for scientific transparency. The updated legends are highlighted in yellow within the revised manuscript for your kind consideration.
Comment 4: If possible, an exploratory multivariable or sensitivity analysis could provide additional perspective, even if it remains outside the main objective of the study.
Authors’ Response: We appreciate this suggestion. While our primary aim was to evaluate the standalone predictive performance of performance status scores, we conducted an exploratory multivariable analysis incorporating relevant clinical covariates in light of your advice. In this model, neither KPS nor ECOG-PS remained independently associated with 90-day survival, as shared below for your kind consideration, further underscoring their limited utility when considered alongside other prognostic factors.
- When controlling for age, ASA class, ambulatory status at presentation, presence of MSCC, SINS score, and PNI, preoperative KPS (OR 1.01 [95 % CI 0.99 to 1.03]; p = 0.23) was found not to independently predict 90-day survival.
- When controlling for age, ASA class, ambulatory status at presentation, presence of MSCC, SINS score, and preoperative albumin level, preoperative ECOG-PS (OR 0.68 [95 % CI 0.42 to 1.09]; p = 0.11) was found not to independently predict 90-day survival.
Reviewer 2 Report
Comments and Suggestions for Authors
The authors evaluated KPS and ECOG-PS predict 90 day survival in a retrospective study of 175 patients at a single institution. They found both measures to have a statistically significant association with 90 day survival, with only fair discrimination and calibration.
Some questions/comments for the authors:
- Are you able to provide more commonly recognized statistical measures, such as sensitivity, specificity, positive predictive value, and negative predictive value? These are often easier for surgeons to conceptualize than discrimination and calibration scores.
- Do patients below a certain threshold of performance status have a <90 day survival? Can you provide a cutoff or is the real-world application so poor that no cutoff exists?
- Was your methodology verified by a statistician?
- Figure 3 makes me believe there is some benefit to utilizing performance scores to predict survival. For example, if KPS is 80 or higher, the chance of 90 day survival is 80% or more. This is compared to <50% chance with a KPS of 20 or 30. ECOG also showed a 90%+ chance of 90 day survival at 0 compared to <50% with an ECOG of 4. While this maybe isn’t perfect, but for a single variable this does seem predictive/important? Can the authors comment as to why they choose to ignore this data and feel performance status is not a clinically significant predictor of survival?
Author Response
Title of the Manuscript: Evaluating the Real-World Predictive Utility of Karnofsky and ECOG Performance Status for 90-Day Survival After Oncologic Surgery for Metastatic Spinal Tumors
Manuscript Number: cancers-3955605
Reviewer 2
Comment 1: The authors evaluated KPS and ECOG-PS predict 90 day survival in a retrospective study of 175 patients at a single institution. They found both measures to have a statistically significant association with 90 day survival, with only fair discrimination and calibration.
Some questions/comments for the authors:
Are you able to provide more commonly recognized statistical measures, such as sensitivity, specificity, positive predictive value, and negative predictive value? These are often easier for surgeons to conceptualize than discrimination and calibration scores.
Authors’ Response: We sincerely thank the reviewer for this excellent suggestion. We agree that traditional diagnostic accuracy measures provide important clinical context that complements our discrimination and calibration analyses. In response, we have added a new subsection (3.2. Diagnostic Accuracy) along with Table 2 to the “Results” section to report the said performance metrics. We have also expanded the “Discussion” section with the following to interpret the findings:
Change to Text: “Our diagnostic accuracy analysis revealed a fundamental challenge in operationalizing performance status for binary surgical decisions, with there being no single cutoff achieving both high sensitivity and specificity. At commonly cited thresholds (KPS ≥ 70, ECOG-PS ≤ 2), sensitivity ranged from 0.66 – 0.76, but specificity remained modest at 0.50 – 0.62, meaning that approximately half of patients who did not survive 90 days would have been classified as acceptable surgical candidates. Conversely, more stringent cutoffs (KPS ≥ 80, ECOG-PS ≤ 1) improved specificity to 0.69 – 0.77 but sacrificed sensitivity to 0.49 – 0.51, potentially excluding many patients who would have survived. This accuracy-coverage tradeoff could elucidate why consensus thresholds remain elusive despite substantial research. Notably, performance status demonstrated the clearest utility at the extremes, with patients having KPS < 40 or ECOG-PS 4 to experience a near-certain 90-day mortality (PPV ≈ 0.75 for mortality prediction), while those with ECOG-PS 0 had an 82% survival probability. However, these unambiguous categories represented a minority of patients in our cohort. The majority fell in intermediate ranges (KPS 40 – 80, ECOG-PS 1 – 3), where diagnostic accuracy was majorly inadequate for confident individual-level prediction. This pattern suggests performance status functions more effectively as a screening tool to identify patients at the margins of surgical candidacy rather than as a precise classifier for the majority falling in the equivocal middle range.”
Comment 2: Do patients below a certain threshold of performance status have a < 90 day survival? Can you provide a cutoff or is the real-world application so poor that no cutoff exists?
Authors’ Response: Thank you very much for this insightful comment. We fully agree that addressing whether a specific performance status cutoff predicts 90-day survival adds important clinical context to our findings. After reviewing the literature, we found that no widely accepted or validated cutoff exists for either the KPS or the ECOG-PS in predicting 90-day survival following metastatic spine tumor surgery. To clarify this point and enhance interpretive context, we have incorporated the following statement in the “Discussion” section. While poorer performance status (such as KPS < 70 or ECOG-PS ≥ 3) is consistently associated with worse short-term outcomes, our study adds to the notion that these thresholds lack sufficient discriminatory accuracy for real-world clinical application.
Change to Text: “However, a widely accepted cut-off exists for neither the KPS nor the ECOG-PS to predict 90-day survival for patients managed with MSTS.”
Comment 3: Was your methodology verified by a statistician?
Authors’ Response: Thank you for this important comment. Although the methodology was not verified by a statistician, both the study methodology and statistical analyses were designed and performed by the primary author, who holds substantial experience in quantitative research design and biostatistical analysis, particularly within the context of clinical outcomes studies in spine oncology. All analyses were conducted using validated statistical methods and were independently reviewed by the senior author to ensure methodological rigor and accuracy.
Comment 4: Figure 3 makes me believe there is some benefit to utilizing performance scores to predict survival. For example, if KPS is 80 or higher, the chance of 90 day survival is 80% or more. This is compared to <50% chance with a KPS of 20 or 30. ECOG also showed a 90%+ chance of 90 day survival at 0 compared to <50% with an ECOG of 4. While this maybe isn’t perfect, but for a single variable this does seem predictive/important? Can the authors comment as to why they choose to ignore this data and feel performance status is not a clinically significant predictor of survival?
Authors’ Response: We sincerely appreciate this thoughtful and perceptive comment. We fully agree that both KPS and ECOG-PS demonstrate meaningful predictive trends at the extremes of functional status—patients with very high KPS or very low ECOG-PS scores generally exhibit excellent short-term survival, whereas those with very poor performance status have markedly reduced survival. Our analysis indeed confirms these associations. Furthermore, the extremes of performance status score also guide surgical decision making, with patients having the upper-most bracket of KPS (and / or the lower-most bracket of ECOG) being good surgical candidates and vice versa.
However, the principal aim of our study was to evaluate the clinical utility of performance status in the context of surgical decision-making, particularly within the intermediate range of predicted survival and performance status strata, where uncertainty is greatest. While KPS and ECOG-PS clearly differentiate outcomes at the extremes, their discriminative ability and net clinical benefit diminish substantially between these boundaries (especially at ≈ KPS 40 – 70 or ECOG-PS 2 – 3). It is precisely within this range that surgical decisions are most challenging and where reliable predictive guidance is most needed. Our findings thus underscore that, although performance status remains an important correlate of survival, it lacks sufficient precision as a standalone predictor to inform nuanced decision-making in real-world MSTS practice. We have updated the “Conclusion” to incorporate the following sentences to reflect this:
Change to Text: “In this single-institution cohort of patients undergoing metastatic spine tumor surgery, preoperative KPS and ECOG-PS were statistically associated with 90-day survival and demonstrated clear clinical relevance at performance extremes. Patients with very high or very low performance status exhibited predictable survival patterns, supporting their utility in straightforward surgical decisions. However, overall predictive performance remained modest, with fair discrimination and calibration. Likewise, the decision-level utility was limited, particularly in the 30 – 60% probability range where clinical ambiguity is most common.”
Reviewer 3 Report
Comments and Suggestions for Authors
This is a very nice study about the prognostic value of Karnofsky index and ECOG on patients with spinal metastasis. The authors have a good structure and make a clear point, using proper methods to reach it.
What would, in my opinion, boost the value of this work would be adding the benefit of those patients analyzed from the surgery. Sure, some survived and some not but how many of them really had benefit from this surgery and in which side of the indexes were those patients who benefited or and those who did not.
Otherwise I am perfectly satisfied with the work done.
Author Response
Title of the Manuscript: Evaluating the Real-World Predictive Utility of Karnofsky and ECOG Performance Status for 90-Day Survival After Oncologic Surgery for Metastatic Spinal Tumors
Manuscript Number: cancers-3955605
Reviewer 3
Comment 1: This is a very nice study about the prognostic value of Karnofsky index and ECOG on patients with spinal metastasis. The authors have a good structure and make a clear point, using proper methods to reach it.
What would, in my opinion, boost the value of this work would be adding the benefit of those patients analyzed from the surgery. Sure, some survived and some not but how many of them really had benefit from this surgery and in which side of the indexes were those patients who benefited or and those who did not.
Authors’ Response: Thank you very much for this thoughtful and constructive comment. We fully agree that evaluating surgical benefit, beyond survival alone, would add substantial clinical value, particularly in understanding which patients derive meaningful functional or quality-of-life improvement from metastatic spine tumor surgery. Although the current study was specifically designed to assess the predictive performance and clinical utility of preoperative performance status (KPS and ECOG-PS) for 90-day survival as a discrete, pragmatic endpoint, we have now acknowledged this as an important direction for future research as follows:
Change to Text: “Additionally, while this study focused on short-term survival as a pragmatic and objective endpoint, it did not assess postoperative functional or quality-of-life benefit. Evaluating which patients derive meaningful improvement from MSTS, particularly across different performance status levels, represents an important next step. Future prospective studies should, therefore, incorporate functional recovery, pain relief, and health-related quality of life alongside survival to better define surgical benefit and optimize patient selection.”
Round 2
Reviewer 2 Report
Comments and Suggestions for Authors
accept
Reviewer 3 Report
Comments and Suggestions for Authors
I am satisfied. Thank you and congratulation.